

# Regional analysis of multivariate compound flooding potential: sensitivity analysis and spatial patterns

Paula Camus[1], Ivan D. Haigh[1], Ahmed Nasr[2], Thomas Wahl[2], Stephen E. Darby[3], Robert J. Nicholls[4]

[1]School of Ocean and Earth Science, National Oceanography Centre Southampton, University of Southampton, Waterfront Campus, European Way, Southampton, SO14 3ZH, UK.
[2]Civil, Environmental, and Construction Engineering & National Center for Integrated Coastal Research, University of Central Florida, 12800 Pegasus Drive, Suite 211, Orlando, FL 32816-2450, USA.
[3]School of Geography and Environmental Sciences, University of Southampton, Avenue Campus, Highfield Road, Southampton, SO17 1BJ, United Kingdom.
[4]Tyndall Centre for Climate Change Research, School of Environmental Sciences, University of East Anglia, Norwich, NR4 7TJ, United Kingdom.

*Correspondence to*: Paula Camus (P.Camus-Brana@soton.ac.uk)

**Abstract.** In coastal regions, floods can arise through a combination of multiple drivers, including direct surface run-off, river discharge, storm surge and waves. In this study, we analyse compound flood potential in Europe caused by these four main flooding sources using state-of-the-art databases with homogenous forcing (i.e., ERA5). First, we perform an analysis to assess the sensitivity of the compound flooding potential to several factors: 1) sampling method; 2) time window to select the concurrent event of the conditioned driver; 3) dependence metrics; 4) wave-driven sea level definition. We observe higher correlation coefficients using annual maxima than peaks over threshold. Regarding the other factors, our results show similar spatial distributions of the compound flooding potential. Second, the dependence between the pairs of drivers using the Kendall's rank correlation coefficient and the joint occurrence are synthesized for coherent patterns of compound flooding potential using a clustering technique. This quantitative multi-driver assessment not only distinguishes where overall compound flooding potential is the highest, but also discriminates which driver combinations are more likely to contribute to compound flooding. We identify hotspots of compound flooding potential located along the southern coast of the North Atlantic Ocean and the northern coast of the Mediterranean Sea.

## 1 Introduction

Floods are the most dangerous and costly natural hazard. For example, in Europe, the economic losses from all natural disasters amounted to EUR 557 billion during 1980-2017 (EEA, 2019), of which 63 % resulted from hydro-meteorological events. Moreover, losses associated with the highest magnitude floods are disproportionately large: around 3 % of these European floods accounted for around 75 % of total deflated losses. The October 2000 flood in Italy and France, for example, was one of the most expensive climate extremes, with damages totalling EUR 13 billion. Since compound floods – floods generated by





different source events occurring concurrently, or in close succession – are often larger than floods generated by an isolated source event, it follows that the adverse consequences of 'compound flooding' are, therefore, likely also disproportionately

large. As an example, the November 1966 coastal flood was one of the most severe observed compound events along the northernmost coast of the Adriatic Sea, which resulted in approximately 25 fatalities and thousand people affected (HANZE database, Paprotny et al., 2018).

The definition of compound events has evolved in recent years. Compound events were defined by the Intergovernmental Panel on Climate Change (Seneviratne et al., 2012) as: "(1) two or more extreme events occurring simultaneously or

successively; (2) combinations of extreme events with underlying conditions that amplify the impact; (3) combinations of events that are not themselves extremes but lead to an extreme event when combined." More recently, Zscheischler et al. (2018) defined compound events as "a combination of multiple drivers and/or hazards that contributes to societal or environmental risk". This new perspective suggests the use of bottom-up approaches to understand the nature of the risks before identifying the relevant drivers and hazards (Zscheischler et al., 2018). However, the impacts of compound events are

commonly felt at a local scale over relatively short timescales, embedded at the same time within larger-scale systems, which requires modelling approaches that fully represent these ranges of space and time scales. To help bridge the gap between the climate science and impact modelling communities, multi-level methodologies which include a quantification of flooding potential using proxies of flood hazard (Bevacqua et al., 2019; Ward et al., 2018; Wahl et al., 2015; Couasnon et al., 2020Ridder et al., 2021) have been used to identify potential hotspots at regional, continental or global scales at a first level. The results

can then be used to inform high-resolution risk assessments, where these are most necessary, which integrate all flooding sources through process-based models to simulate their physical interaction (Bevacqua et al., 2019) as well as their interaction with human systems (Sebastian et al., 2019, Wang et al., 2020).

According to the proposed typology in Zscheischler et al. (2020b), flooding is considered as a multivariate event because multiple climate drivers and/or hazards can occur in the same geographical region, that may not be extreme themselves, but

their joint occurrence causes an extreme impact. In coastal regions, flooding can arise from the combination of multiple sources: pluvial (direct surface runoff), fluvial (increased river discharge) and/or oceanographic (storm surges plus tides and/or waves). The main drivers of flooding are typically causally related through their associated weather patterns, for instance, when a storm causes extreme rainfall and/or a storm surge and/or high waves. The statistical modelling approach suggested for this typology consists of multivariate probability distribution functions, which represent both the marginal and joint features

of multiple random variables (Zscheischler et al., 2020b). High-dimensional systems can be modelled using copula-based approaches, but due to their complexity these multivariate statistical models have been limited to local scale studies (Bevacqua et al., 2017, Couasnon et al., 2018). At global or regional scales, where compound flooding risk varies substantially along coastlines, the risk is estimated indirectly by quantifying the dependence limited to bivariate drivers (proxies of the flooding hazard). For example: 1) Zheng et al. (2013) found a significant dependence between maximum daily storm surge and daily

precipitation along the coast of Australia; 2) Wahl et al (2015) detected an increasing risk of compound flooding from storm surge and precipitation for major U.S. coastal cities, and 3) Hendry et al. (2019) analysed the characteristics of compound





flooding arising from the combination of river discharge and sea level along the UK coast. Such quantifications of compound flooding potential are based on dependence measures (e.g., using correlation coefficients, Wahl et al., 2015; or joint occurrence, Hendry et al., 2019) or bivariate statistical models (e.g., bivariate logistic threshold-excess model, Zheng et al., 2013; or

copulas, Wahl et al., 2015, Moftakhari et al., 2017, Ward et al., 2018). Furthermore, recent advances in large-scale sea level and river discharge modelling (Muis et al., 2016, Yamazaki et al., 2014) which provide time-series of these drivers over durations of more than 30 years, allow the identification of potential hotspots at country, continental and global scales (Wu et al., 2018; Bevacqua et al., 2019, 2020; Couasnon et al., 2020).

Regarding the identification of compound events, conditional sampling is usually applied (Wahl et al., 2015; Ward et al., 2018;

Couasnon et al., 2020) which implies that compound events are conditioned to one of the drivers being extreme. For this reason, when limiting to a bivariate characterization of compound events (e.g., when using correlation coefficients), two subsets of extreme events are identified, and the dependence is analysed when one or the other of the variables is extreme. Another option is to select pairs of high values when both variables exceed individual high percentiles (e.g., 95th percentile, as Bevacqua et al., 2019), but in this case, compound events are defined only when both individual drivers are characterized

as being extreme. This issue is similar to what happens when measuring compound flooding potential based on factors or indices that quantify the effect of the dependency using copulas with AND hazard scenarios (Ward et al., 2018, Ganguli and Merz, 2019, Couasnon et al., 2020). However, it could be sufficient that only one of the driver variables was extreme to make a bivariate occurrence hazardous (OR hazard scenario, Moftakhari et al., 2017).

In the compound flooding studies summarised above, it is evident that a wide range of different statistical approaches have

been used to define compound flooding potential, usually caused by the combination of two drivers. Only Hendry et al. (2021, submitted) have to date considered all four potential drivers of flooding in coastal regions (precipitation, river discharge, sea level and waves). Recently, Ridder et al., 2021 have identified hotspots of compound events that potentially cause high-impact floods related to wet conditions based on the joint occurrence of multiple hazards pairs (precipitation, wind, hail, streamflow and storm surge). In other studies, the wave component has typically been included in the sea level directly (by combining

wave height with storm surge and/or astronomical tide) without analysing the correlation with the other drivers (Bevacqua et al., 2019). Similarly precipitation and river discharge have often been considered as an equivalent driver in the analysis of compound coastal flooding in combination with storm surge (Bevacqua et al., 2020). Additionally, different sampling methods to identify compound events and dependence measures to quantify compound flooding potential are used. The net effect of these variations in practice is to complicate comparisons between different studies.

The overall aim of this paper, is to perform a regional analysis along the North Atlantic, Mediterranean and Black Sea coastlines of the compound flood potential caused by pluvial, fluvial and oceanographic sources during 1979-2018, using state-of-the-art model hindcasts with homogenous forcing (i.e., ERA5). In addition to this aim, two specific objectives are defined: 1) to assess the sensitivity of the compound flood potential to several factors that can affect the identification of compound events and the analysis of the spatial distribution of compound flooding potential, and 2) to detect different types of compound

events and spatial patterns of compound flooding potential that arise from the combination of the four drivers. The paper is





structured as follows. The datasets and methods are detailed in Section 2s and 3, respectively. The results of the two objectives are discussed in Section 4. Key findings are discussed in Section 5, with conclusions given in Section 6.

**2 Data**

We use modelled data to cover the entire coastlines that are focus of this study, using long-term, spatially continuous and
temporally consistent gridded data for all four flood source variables, as discussed in each of the sub-sections below. Of note is that although these databases are not available on a common grid and there are differences in their spatial resolution, they are all derived from the European Centre of Medium-Range Weather Forecast (ECMWF)'s latest global atmospheric reanalysis (ERA5, Hersbach et al., 2020), or from hindcasts forced by this atmospheric reanalysis. The new database GloFAS-ERA (Harrigan et al., 2020) is used for the first time to characterize river discharge when studying compound flooding potential.
We do not account for pluvial flooding directly, as pluvial flooding is a much smaller scale process. Instead, following Wahl et al (2015), we use precipitation at each analysis site as a proxy for surface runoff potential. Our four flood source variables are, therefore: precipitation (P), river discharge (Q), storm surge (S) and waves (W), this latter variable being characterized by the significant wave height. Each of the four databases employed are described in the following sub-sections, including how we have selected the locations for the sensitivity analyses and compound event characterization.

**2.1 Precipitation time-series**

Precipitation time-series have been extracted from the ERA5 reanalysis, which is based on the Integrated Forecasting System (IFS) Cy41r2, which has been used in the ECMWF operational medium range forecasting system since 2016 and benefits from a decade of developments in model physics, core dynamics and data assimilation (Hersbach et al., 2020). The ERA5 reanalysis replaces the ERA-Interim reanalysis with a significantly enhanced horizontal resolution of 31 km, compared to 80 km for
ERA-Interim. Long-term (1998-2018) and monthly average precipitation rates from ERA-Interim and ERA5 have been evaluated by comparing them with values from other datasets (e.g., the version 7 of NASA's TRMM Multi-satellite Precipitation Analysis (TMPA) 3B43 dataset, Huffman et al. 2010) and there is a marked improvement in the estimated precipitation in ERA5 compared to ERA-Interim (Hersbach et al., 2020). The ERA5 hourly dataset spans 1979 onwards and it is currently publicly available at the Copernicus Climate Change Service. Here, accumulated daily precipitation is calculated
from hourly data.

**2.2 River discharge time-series**

River discharge time-series were extracted from the Global Flood Awareness System (GloFAS)-ERA5 reanalysis (Harrigan et al., 2020). This reanalysis is a global gridded dataset (excluding Antarctica), with a horizontal resolution of 0.1° at a daily time step and with a 40 years long duration starting 1 January 1979. The GloFAS-ERA5 river discharge reanalysis was
produced by coupling the land surface model runoff component of the ECMWF ERA5 global reanalysis with the LISFLOOD



hydrological and channel routing model (van der Knijff et al., 2010). LISFLOOD allows the lateral connectivity of ERA5 runoff grid cells routed through the river channel to produce river discharge. ERA5 runoff is produced from the HTESSEL land surface model (Hydrology Tiled ECMWF Scheme for Surface Exchanges over Land; Balsamo et al., 2009) with an advanced land data assimilation system to assimilate conventional in-situ and satellite observations for land surface variables.

Groundwater and river routing parameters in GloFAS were calibrated against river discharge observations for 1,287 catchments globally by Hirpa et al. (2018). A total of 463 of the largest lakes (surface area > 100 km2) and 667 largest reservoirs have been incorporated into the GloFAS river network.

### 2.3 Storm surge time series

Hourly and daily storm surge time-series have been extracted from the Coastal Dataset for the Evaluation of Climate Impact
(CoDEC) (Muis et al., 2020). The third generation Global Tide and Surge Model (GTSM, Kernkamp et al., 2011), with a coastal resolution of 2.5 km (1.25 km in Europe), was forced with meteorological fields from the ERA5 climate reanalysis to simulate extreme sea levels for the period 1979 to 2017. Besides the increase of the resolution of GTSM v3.0 from 5 km along the coast (50 km in the deep ocean) to 2.5 km along the coast (25 km in the deep ocean), the GTSM v3.0 model performance for tides was also improved by the implementation of additional physical processes. The validation against observed sea levels
demonstrated a good performance which reflects not only the more skilful hydrodynamic simulations but also the accuracy of the meteorological forcing. ERA5 represents better the evolution of weather systems due to an increase of the spatial and temporal resolution. The annual maxima had a mean bias 50% lower than the mean bias of the previous Global Tide and Surge reanalysis dataset (Muis et al., 2016).

### 2.4 Wave time series

Hourly wave time-series have been extracted from the ERA5 reanalysis at a spatial resolution of 0.5° with some improvements and updates compared to ERA-Interim (Hersbach et al., 2020). The model bathymetry was updated to use a more recent version of ETOPO2 (NOAA 2006). A new wave advection scheme was introduced in the WAM model with a revised unresolved bathymetry scheme to better account for the propagation along coastlines and to better model the impact of unresolved islands (Bidlot 2012). The slow attenuation of long period swells as well as the impact of shallow water on the wind input was
introduced with an overall retuning of the level of dissipation due to white-capping (Bidlot 2012). The atmosphere and ocean are coupled by a two-way interaction: atmosphere generates ocean waves through the surface wind stress, while the waves influence the atmospheric boundary layer via sea-state dependencies in the surface roughness. Altimeter measurements were used to assimilate information on significant wave height. Independent buoy data were used for validation showing significant improvement in the wave height in ERA5 data compared to ERA-Interim (Hersbach et al., 2020).



### 2.5 Selection of the study locations

The spatial resolution of the four datasets is shown in Figure 1a for the area of Ireland, UK and north-western France. The ERA5 precipitation database has a resolution of 0.25°x0.25°, the ERA5 wave database has a resolution of 0.5°x0.5°, the CODEC storm surge database has a resolution of 2.5 km along the coast, and the GloFAS-ERA5 river discharge database has

a resolution of 0.1°x0.1°. The river network data implemented in GloFAS for routing operations was produced using fine-scale hydrography inputs from HydroSHEDS (Lehner et al., 2008).

We used HydroSHEDS data to identify the mouths of rivers with a catchment area higher than 1,000 km2 to adjust the distribution of study locations for a regional analysis. The GloFAS grid nodes closest to the locations of the river mouths were checked and we selected the grid node with the highest river discharge. The closest grid nodes of the precipitation, wave and

storm surge databases to the selected GloFAS grid nodes were identified and are shown in Figure 1b. Across the whole domain (see Figure S1) we analyse 540 locations.

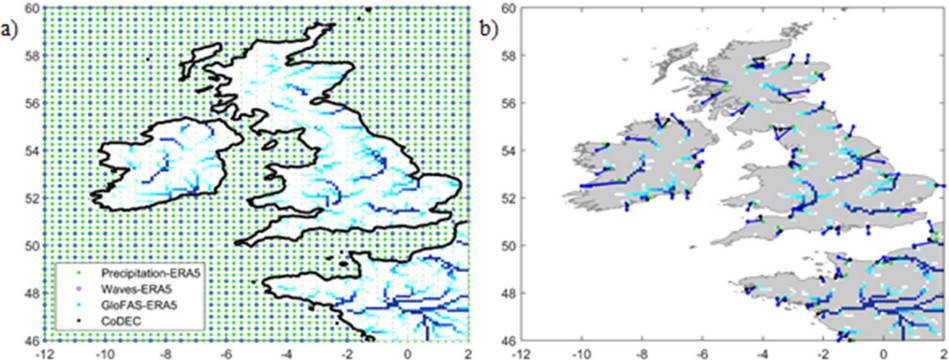

**Figure 1: a) Spatial resolution of the precipitation (ERA5), river discharge (GloFAS-ERA5), storm surge (CoDEC) and wave (ERA5) data in the area of Ireland, UK and north-western France. b) Selected river discharge grid nodes at the mouth of rivers with a**
**catchment area > 1000 km2 and the closest precipitation, storm surge and wave grid nodes along these coasts.**

### 3 Methods

We characterize the compound flooding potential generated by the four flood drivers (P, Q, S, W) by calculating the dependence between all possible pairs of the four main source variables along the coasts of the Eastern North Atlantic, Mediterranean Sea and Black Sea. In addition, we also superimpose linearly the storm surge and wave components into a

combined sea level, ignoring the astronomical tidal component of sea level, as it is deterministic. We do this considering two definitions of the wave-driven sea level component: 1) a simplified for the wave contribution to sea level (e.g., Vousdoukas et al., 2017), called this variable SW, and, 2) through the use of semi-empirical formulations (e.g., Vitousek et al., 2017), that also take into account the wave period (Tp) to calculate Setup. We term to this second definition of sea level, as the sum of S





and Setup, water level (WL) and compare results of compound flooding potential using both definitions. The seven paired
driver combinations we consider here are, therefore: 1) Q-P [P-Q]; 2) Q-S [S-Q]; 3) Q-W [W-Q]; 4) Q-SW [SW-Q]; 5) P-S
[S-Q]; 6) P-W [W-P], and7) P-SW [SW-P]. Here the indicated leading variable is the conditioning variable (dominant driver)
and the second one is conditioned on the first (secondary driver) in the two-sided conditional sampling we apply.

In the following sub-sections, we first describe the sensitivity analysis designed to elucidate the extent to which the
methodology employed affects the quantification of compound flooding potential. Second, we describe the clustering method
used here to identify different types of multivariate compound events and the spatial distribution of compound flooding
potential thereby arising across the four flood drivers considered in this study.

### 3.1 Sensitivity analysis

Our first objective is to assess the sensitivity of compound flood potential to different aspects of the underpinning methodology:
1) sampling method used, 2) time window investigated, 3) dependence metrics employed, and, 4) definition of wave-driven
sea level. Each of these sensitivity tests is discussed in turn below.

### 3.1.1 Sampling

Evaluation of the dependence between multiple drivers is limited to a bivariate analysis which imposes a two-sided conditional
sampling to select multivariate extremes. Compound extreme events are defined in previously published studies are either
selected using Annual Maxima (AM) or Peak Over Threshold (POT). Although Ward et al. (2018) performed a sensitivity
analysis and compared correlations between river discharge and storm surge using both POT with two thresholds (equal to
95th and 99th percentile) and AM sampling, here we build on that prior work to examine also the effect of the sampling method
in the dependence between the four coastal flooding drivers considered. The disadvantage of the AM is that events are selected
that might not be considered extreme in the dominant variable. On the other hand, the POT approach increases the amount of
selected extreme events but introduces two parameters in the selection process: 1) the definition of the time between peaks for
each peak to be considered an independent event, and; 2) the threshold. The independence between extreme events is assured
by declustering the events based on the duration of the storms in the study area and selecting the highest event within each
storm. The criteria used to select independent events in this study comprise a storm duration of 5 days for river discharge and
3 days for precipitation, storm surge and waves. These values were selected based on an analysis of the duration of highest
storms conditioned to each variable in the study domain and following numbers used in previous studies (Ward et al., 2018,
210 Hendry et al., 2019, Marcos et al., 2019, Bevacqua et al., 2019). Many methodologies for an automated threshold selection
have been proposed based on graphical methods combined with goodness of fit (e.g., Solari et al., 2017) but such techniques
are difficult to implement in regional studies due to the different characteristics of time series of the several drivers involved
in coastal flooding. Hence, we decide to apply the POT method with a threshold that guarantees 3 (POT3) or 6 (POT6)
events/year to analyse also the effect of the value of the threshold.





### 3.1.2 Time window

The conditional sampling introduces another factor that could affect the definition of compound events and this is related to the selection of the concurrent value of the secondary variable to the identified extreme events of the dominant variable. Specifically, there could be a temporal lag between variables that leads to a potential coastal flooding event. This lag can be implemented after identifying both series of extremes from the two drivers (Hendry et al., 2019) or by a time window when identifying the value of the conditioned variable (Wahl et al., 2015, Ward et al., 2018, Coausnon et al., 2019). Once a time window ($\Delta t$) is established, the value of the secondary variable is selected as the maximum value within $\pm \Delta t$ days. A variety of temporal windows have been considered, from zero lag (Bevacqua et al., 2020), through $\pm 1$ day (Wahl et al., 2015), $\pm 3$ days (Couasnon et al., 2020) to $\pm 5$ days (Ward et al., 2018, Hendry et al., 2019). Furthermore, although river discharge data have been extracted at the river mouth, not all the databases for the four coastal flooding drivers have the same spatial resolution, varying considerably the distance between grid nodes at each location of the study domain. Therefore, here we test the sensitivity of the identification of bivariate compound events to time windows between 1 and 10 days or 1 and 3 days, keeping the lag which provides the highest correlation coefficient between each pair of variables at each location.

### 3.1.3 Dependence metrics

Several correlation coefficient definitions (e.g., Kendall's tau, Wahl et al., 2015; or Spearman's rho, Couasnon et al., 2020) and other metrics for the characterization of the dependence between events when both drivers are extreme (e.g., joint occurrence, Hendry et al., 2019, or tail dependence, Marcos et al., 2019) have been used in previous studies. Here we analyse the extent to which characterization of compound events could be affected by the selection of these varying dependence metrics.

**Correlation coefficient**: We use the Kendall's rank correlation coefficient tau and Spearman's rank correlation coefficient rho because they are commonly used nonparametric methods of detecting associations between two variables. Significance is assessed at $\alpha = 0.05$ (i.e., 95% confidence level), using corresponding p values. The Spearman's rank correlation between two variables is defined as the covariance of the two variables normalized by the product of their standard deviations between the rank scores of those two variables. The Kendall's tau is also a rank order correlation coefficient which quantifies the difference between the % of concordant and discordant pairs among all possible pairwise events. The Kendall's correlation is considered to be more robust than the Spearman's correlation because it offers better estimates with smaller asymptotic variance and is less susceptible to outliers (Ganguli and Merz, 2019).

**Joint occurrence**: The 'joint occurrence method' (Hendry et al., 2019) consists of simply counting the number of times extreme events are identified in the two drivers analysed within the time window ($\Delta t$) considered.

**Tail dependence**: The dependence structure in the tails between two variables can be measured by the pair of statistics ($\chi, \bar{\chi}$) (e.g., Coles et al., 2001). Both coefficients $\chi$ and $\bar{\chi}$ are defined as limit values which tend to 1 if both variables are asymptotically dependent over a certain threshold. The coefficient $\chi$ represents the probability of bivariate extreme events




when both variables are extreme and provides a measure of the dependence strength (Marcos et al., 2019), referred to as extremal correlation (Zscheischler et al., 2020a). $\bar{\chi}$ is the residual tail dependence coefficient and contains additional information about the association ($-1 < \bar{\chi} < 1$) between extremes of both variables when they are asymptotically independent

($\chi=0$). We use the function taildep from the R package extRemes (Gilleland and Katz, 2016) to derive these values.

### 3.1.4 Definition of wave-driven sea level

Although non-linear interactions between storm surges and waves could amplify the magnitude of the sea level, the assumption that both contributions may be linearly summed is generally adopted and has often been used as a proxy of coastal flooding driven by oceanographic variables (Rueda et al., 2016, Bevacqua et al., 2019, Marcos et al., 2019). Regarding the wave

contribution to sea level, when wind-generated waves approach nearshore and break in the shallow surf zone, they induce variations of the sea level at different time and space scales enhancing coastal flooding. The highest wave-driven contribution to the total water level, called run-up, depends on two dynamically different processes: (1) wave setup, which is a time-average sea level rise occurring over a few hours to several days, and which is determined by local wind sea and swell conditions, and (2) swash which is a high-frequency process by which sea level fluctuates due to individual incident waves, with an additional

low-frequency component generated by infragravity waves (related to the presence of groups in incident short waves). The magnitude and expanse of both components depends on the sea-state characteristics (significant wave height, period and spectrum shape; Guza and Fedderson, 2012), as well as nearshore bathymetry and topography. Spatially, setup could extend from tens of meters in steep coastal areas to several kilometres in low-sloping coastal areas, while runup extension varies from a few meters to on the order of a hundred meters in reflective and dissipative environments, respectively (Dodet et al., 2019).

Runup is not usually included in the wave component of the sea level driver in coastal flooding analysis because its temporal duration is on the order of hours and requires local geological characteristics that could artificially inflate the wave contribution in global and regional studies (Aucan et al., 2018). The setup contribution is defined using empirical formulations with different levels of sophistication. Wave setup has been approximated as the significant wave height multiplied by 0.2 (Vousdoukas et al., 2018, Bevacqua et al., 2019, Marcos et al., 2019) or by applying the Stockdon formulation (Stockdon et al., 2006) with

different parameterizations (Vitousek et al., 2017, Rueda et al., 2017, Melet et al., 2018). The wave setup contribution to the total water level is very sensitive to this parameterization (Aucan et al., 2018). Furthermore, in the definition of the sea level as the sum of S and 0.2W, here called SW, we have also applied the Stockdon formulation for wave setup for dissipative beaches (as Vitousek et al., 2017) which is known to provide similar results as using a beach slope of 0.02 (~50% of the world's beaches have slopes smaller than 0.02, Aucan et al., 2018), here termed water level (WL), as the sum of S and Setup.

### 3.2 Characterization

The evaluation of compound flooding potential due to the combination of four drivers based on a bivariate analysis is a complex problem due to the high dimensionality (e.g., spatial variability of the dependence metrics and relative contribution of each pair of drivers). We apply a two-step cascade classification with two sub-objectives: 1) to analyse the dependence between the





metrics that characterize the bivariate compound flooding potential between the pairs of drivers, and 2) to extract spatial

patterns from this dataset in order to identify hot spots of compound flooding potential arising from P-Q-S-W.

The two-step classification method consists of the use of self-organizing maps (SOM) as the first step and applying the k-means algorithm (KMA) as the second step (Rueda et al., 2017). In this study, SOM is applied first to take advantage of the powerful visualization characteristics but not to obtain a reduced number of clusters. The k-means algorithm is a classification technique that divides the high-dimensional data space into a number of clusters, each one defined by a centroid and formed

by the data for which the centroid is the nearest (Hastie et al., 2001). The SOM automatically extract clusters of high-dimensional data and projects them into a two-dimensional organized space (2D lattice), allowing an intuitive visualization of the classification (Kohonen, 2000). A SOM algorithm is a version of the KMA in which the centroids are forced with a neighborhood adaptation mechanism to a 2D lattice preserving the original topology of the data and producing that similar patterns in the original space are close in the 2D lattice. The Maximum-Dissimilarity-Algorithm (MDA, Camus et al., 2011)

is applied to initialize the KMA to obtain a better distribution of the centroids over the multi-dimensional data space and avoid random initialization. The optimal number of clusters is evaluated using the Davis-Boudin (Davies and Bouldin, 1979) and the gap criteria (Tibshirani et al., 2001).

## 4 Results

### 4.1 Sensitivity analysis

This section describes the results for the first objective, relating to the sensitivity analysis. Results from each of the four sensitivity tests is described in turn.

### 4.1.1 Sampling

Two-sided conditional sampling has been applied to the seven pairs of drivers identified when applying the AM and the POT methods with either 3 or 6 events per year. Figure 2 shows the comparison of the correlation coefficients between Q and P

using the three approaches when either Q or P are the dominant drivers. Only this pair of variables is shown because it presents the highest correlation and the purpose of this subsection is only to test the sampling method. Locations are divided in five regions (see Figure S1 for the locations of these regions): Northwestern Europe (NEW, 99 locations); Northeastern Europe (NEE, 165 locations), Southern North Atlantic coast (SNA, 60 locations), Western Mediterranean Sea (WM, 99 locations), and Eastern Mediterranean Sea (EM, 117 locations). Correlation is higher between the conditional pairs of extremes selected

using AM while similar correlation is obtained using POT3 or POT6, with higher dispersion in the lower values of the coefficients. Correlation coefficients calculated with AM subsets of extremes are, on average, around 0.2 higher than those derived with the POT approach and this is consistent across all regions. Scatter plots display data only for those locations where significant ($p<0.05$) correlation coefficients are present. Overall, conclusions about the comparisons for all other pairs of drivers are similar to those for Q-P (Figure S2).




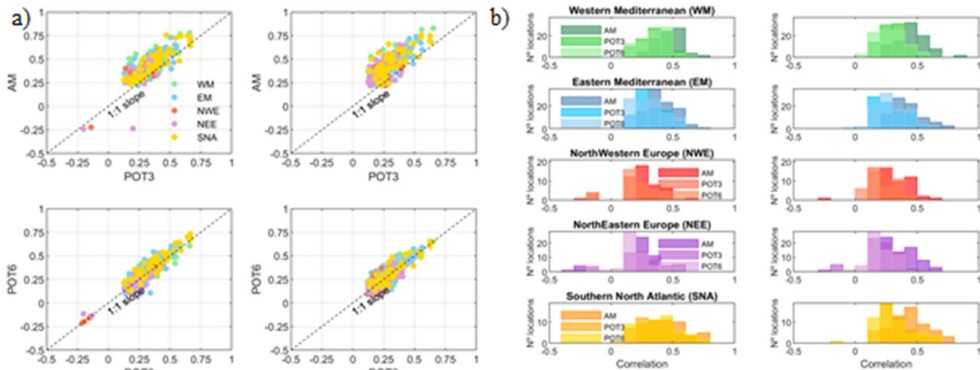

**Figure 2: a) Scatter plots of the Kendall's correlation (p< 0.05) between Q and P using POT3 vs AM (upper panels) and POT3 vs POT6 (lower panels), using either Q (left panels), or P (right panels) as the dominant variable. b) Histograms of the correlation coefficients using the three approaches (AM, POT3 and POT6) for each region when either Q (left panels) or P (right panels) as the**
**dominant drivers.**

Figure 3 shows the comparison of the number of co-occurring events between Q and P using the three approaches. The number of events is higher when using POT3 compared to AM and when using POT3 compared to POT6, as expected. The scatter plots follow roughly the 1:3 or 1:2 slope, respectively, indicating an approximate tripling or doubling in the number of events
between different approaches. However, the spatial pattern is similar (correlation coefficient between the number of co-occurring events is around 0.93-0.98) which means that the three methods identify broadly equivalent areas where prone to compound events with both variables being extreme. Results for all pairs of drivers are shown in Figure S3, which has a similar behaviour, albeit with an overall lower number of co-occurring events.


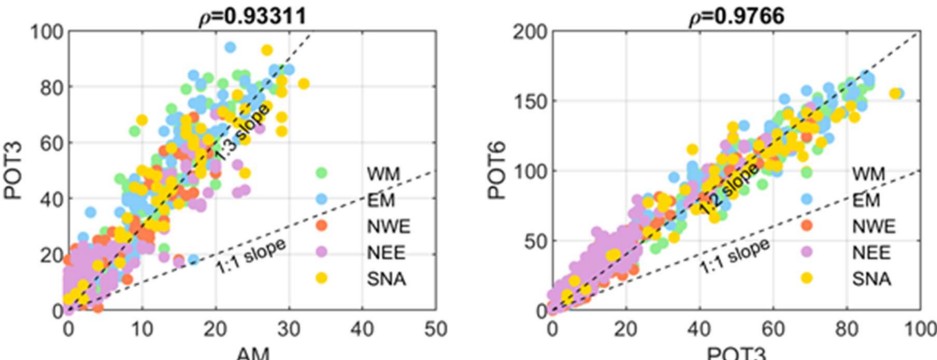

**Figure 3: a) Scatter plots of the number of co-occurring events between Q and P using AM vs POT3 (left panel) and using POT3 vs POT6 (right panel). Dashed lines mark relationships between AM and POT3 or between POT3 and POT6 covering scaling factors equal to 2.0 and 3.0.**


### 4.1.2 Time window


Figure 4 shows the comparison between the highest correlation coefficient obtained when using a time window ($\Delta t$) of ± 3 days versus using $\Delta t$ = ± 10 days for the pairs of variables Q and P, S, W or SW using POT3, and the join occurrence between Q and P, S, W or SW. Only locations with a significant correlation ($p < 0.05$) are represented in Figure 4. Results indicate there is no major difference in the correlation between drivers when employing the two investigated time windows. Larger

differences (~0.1 higher correlation) are obtained when Q is the dominant variable in few locations (9 of the 540) where correlation is moderate overall (~0.30). The joint occurrences tend to be slightly higher (10 number of co-occurring events) with a higher time window but also fewer compound events (by half) are detected in locations with low-medium join occurrence (<~40) using a $\Delta t$ = ± 3 days (lower left corner of scatter plots in Figure 4b).



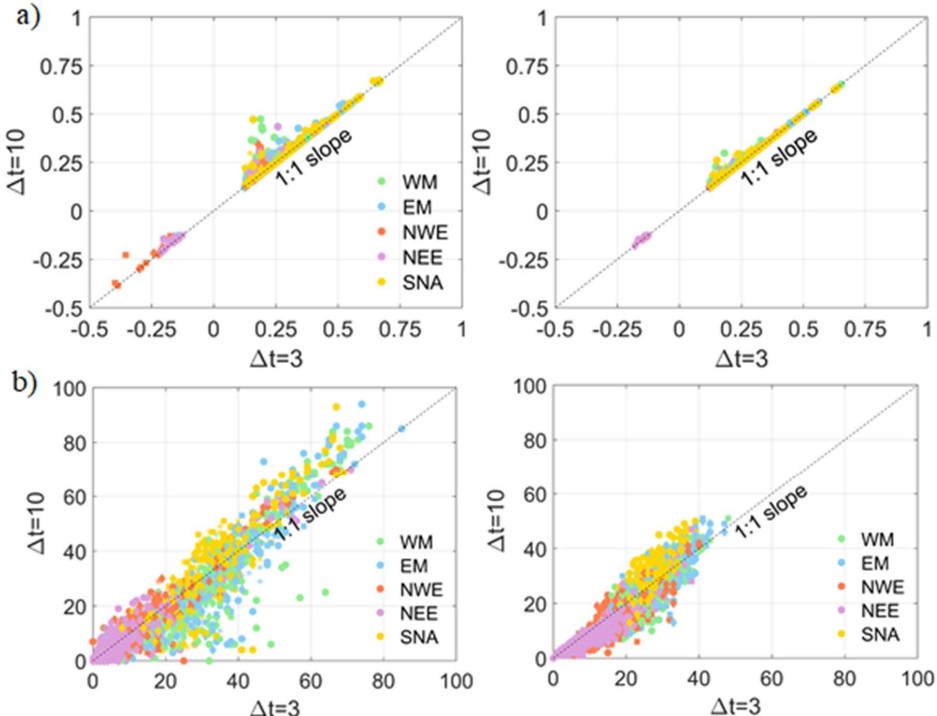


**Figure 4: a) Scatter plots of the highest correlation coefficient (p< 0.05) between Q and P (●), S (✳), W (■) and SW (♦) using a time window (Δt) of ± 3 days and 10 days with a POT3 approach, using Q either as the dominant (left panel) or secondary (right panel) variable in the conditional sampling. b) Scatter plots of the joint occurrences between Q and S, W and SW (left panel) and between P and S, W and SW (right panel); markers are the same as in (a).**


### 4.1.3 Dependence metrics

Figure 5a shows the comparisons when using the Kendall's versus Spearman's rank coefficients for the pairs of variables (Q-P/P-Q, Q-S/S-Q, Q-W/W-Q, Q-SW/SW-Q, P-S/S-P, P-W/W-P, P-SW/SW-P), using a time window of ± 3 days and across the three approaches (AM, POT3, POT6) considered in the conditional sampling. There is a categorical correspondence between

both correlation coefficients, with Kendall's coefficient having a tendency to be smaller than the Spearman's coefficient. Therefore, to characterize compound events in terms of correlation and its spatial distribution, the information provided by both coefficients is equivalent. The number of locations with significant correlation is very similar with both correlation coefficients (see Table 1).


Regarding $(\chi,\bar{\chi})$ statistics, the usual way to decide the threshold involves making a visual examination of the evolution of their

empirical estimates for increasing threshold levels (Zscheischler et al., 2020a). Here, we decided to estimate χ at a probability

threshold 0.95 after careful examination of the results for different levels. The comparison between χ and the joint occurrences

divided by 3 events per year and the number of years (40) for the pair of variables Q-S is shown in Figure 5b. There is high

correspondence (correlation coefficient around 0.9) between both dependence metrics mainly because both of them measure

the probability of bivariate extremes when both drivers are extreme.  The remaining small differences may be due to different

sampling processes leading to different extreme subsets. The statistic χ is estimated using the empirical distribution of the daily

time series of Q (mean daily values) and S (maximum daily values) and the extremes are selected without any clustering. On

the other hand, the number of co-occurring events was calculated using a POT with a threshold that guarantees 3 events per

year and with a storm duration of 3 or 5 days to select independent events. A similar relationship between both metrics has

been found for other combinations of variable pairs (not shown here).


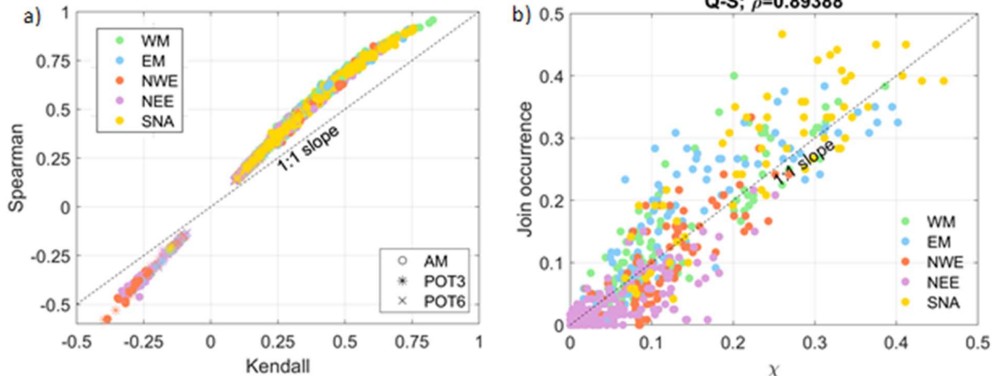

**Figure 5: a) Scatter plot of the Kendall's correlation coefficient (p< 0.05) vs Spearman's correlation coefficient (p< 0.05) between all pairs of variables. b) Scatter plot of the statistic χ (threshold = 0.95) against the joint occurrence divided by 3 events per year for the pair of variables Q-S.**


| Dominant variable | Kendall's tau | | | | | | Spearman's rho | | | | | |
|---|---|---|---|---|---|---|---|---|---|---|---|---|
| | Variable1 | | | Variable2 | | | Variable1 | | | Variable2 | | |
| Pairs of variables | AM | POT3 | POT6 | AM | POT3 | POT6 | AM | POT3 | POT6 | AM | POT3 | POT6 |
| Q-P | 307 | 301 | 334 | 332 | 327 | 355 | 309 | 306 | 339 | 332 | 325 | 353 |
| Q-S | 101 | 145 | 212 | 126 | 137 | 204 | 105 | 144 | 213 | 126 | 141 | 204 |
| Q-W | 116 | 166 | 262 | 0 | 122 | 193 | 114 | 171 | 260 | 0 | 116 | 195 |
| Q-SW | 124 | 178 | 251 | 0 | 139 | 210 | 120 | 182 | 254 | 0 | 146 | 213 |
| P-S | 47 | 69 | 141 | 124 | 173 | 281 | 47 | 69 | 139 | 120 | 171 | 279 |
| P-W | 87 | 123 | 204 | 0 | 214 | 363 | 87 | 122 | 203 | 0 | 219 | 360 |
| P-SW | 83 | 128 | 210 | 0 | 223 | 353 | 83 | 126 | 208 | 0 | 222 | 355 |

**Table 1: Number of locations with significant correlation (p< 0.05) being 540 the total number of locations.**


### 4.1.4 Definition of wave-driven sea level

The effect of the definition of sea level (oceanographic flooding drivers) in the characterization of compound events is analysed here by comparing the Kendall's correlation coefficients obtained between the pair of variables Q or P when using the two varying sea level definitions (SW and WL) used in this study (Figure 6). Differences in the obtained correlation coefficients are typically small (mainly around 0.05), except along the southern Atlantic coast where the differences are slightly higher than 0.15. The southern Atlantic coast region presents the highest correlations between pluvial or fluvial sources and

oceanographic drivers (correlation coefficients around 0.6-0.7), within the entire study domain meaning that the identification of this region as an area with significance dependence is still preserved. The effect of the sea level definition on the correlation when Q or P are the dominant variables in the conditional sampling is much smaller (around 0.05, not shown here).

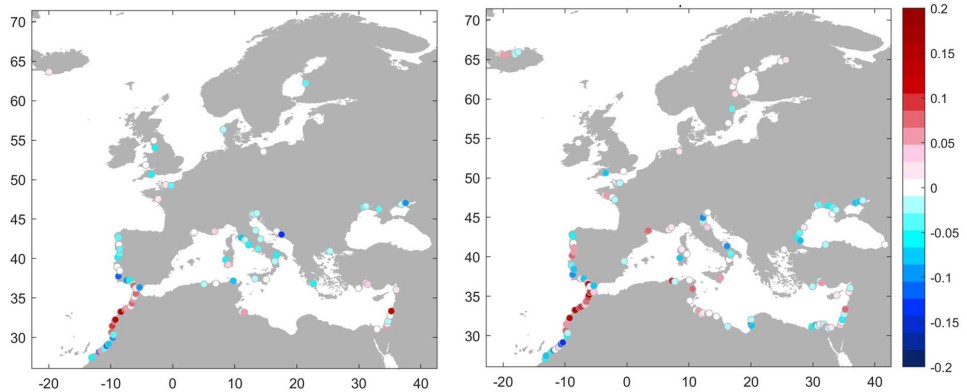

**Figure 6: Differences in the Kendall's correlation coefficient (p<0.05) obtained when using Q and SW and Q and WL (left panel) and P and SW and P and WL (right panel) when SW or WL are the dominant variables.**

### 4.2 Characterization of compound flooding potential

This section describes the results obtained in relation to the second objective and contains: 1) a description of the dependence

between all identified pairs of drivers based on the Kendall's correlation coefficient and the joint occurrences, 2) formulation of a severity index to represent the metocean climate in the study domain and which combines the extremeness of the four drivers, and 3) an identification of spatial patterns of compound flooding potential derived based on a classification of the dependence metrics between the pairs of drivers and the severity index.





### 4.2.1 Dependence between pairs of the four flooding drivers


Here we consider the results obtained using the POT3 method and a time window of ± 3 days to characterize the multivariate compound events along the North Atlantic, Mediterranean and Black Sea coasts. The analysis of the dependence between S-W is performed at an hourly temporal resolution and using a smaller Δt (1 day) because it is considered that both drivers contribute simultaneously to the definition of sea level. The dependence between S-W as calculated using daily data is

compared with the results using hourly data (Figure S4). A reduction of the correlation coefficients in the whole study domain, especially along the Atlantic coasts of Spain and France and along Baltic Sea coast is evident while the number of joint occurrences increases in almost all locations, especially along the Atlantic coast of the Iberian Peninsula up to North of Africa and in the Mediterranean Sea.

Kendall's correlation coefficient and the joint occurrences between the four pairs of variables: Q-P, Q-S, Q-W, and Q-SW are

represented in Figure 7. The variables Q and P (Figure 7-a) present correlation coefficients of around 0.6-0.7 along the most southern coasts of the Atlantic study region and also in some locations in the Mediterranean Sea (coast of Gibraltar Strait, Algeria, southern Italy, east coast of Turkey and Levante Region in the eastern Mediterranean), while correlation coefficients of around 0.1-0.2 are more predominant along northern European coasts (except the west coast of Jutland and west coast of the UK). Similar spatial correlations are found whether Q (red scale) or P (blue scale) are used as the dominant variable in the

conditional sampling, except in locations along the French coast of the English Channel, the eastern coast of UK, and the coasts of Tunisia and Libya, where higher correlations are obtained when the compound events are conditioned to P. The joint occurrence (circle size) presents a similar spatial pattern to the correlation coefficient between these two variables, with the maximum number of co-occurring events close to 100.

For Q and S (Figure 7-b), highest correlations are of around 0.3-0.4 when both drivers are dominant, mainly along the southern

Atlantic coasts of the Iberian Peninsula, north of Africa and Gibraltar Strait. The dependency is slightly higher when Q is the main variable in the identification of compound events, even the only correlation along most of the northern coasts of the Mediterranean Sea. Higher joint occurrences are detected in locations with higher correlation, with around 50-60 of co-occurring events. However, similar numbers of joint occurrences are found in locations along the eastern coast of Italy and the eastern north Mediterranean coasts with lower correlation.

For Q and W (Figure 7-c), the spatial distribution is quite similar to Q-S with slightly smaller correlation along the most southern Atlantic coast and higher along the west coast of Iberian Peninsula. Correlation only when Q is the dominant variable is even more pronounced in locations along the Mediterranean Sea (e.g., eastern coast of Spain). These spatial patterns are also reflected in the distribution of the correlation coefficient between Q and SW as a combination of both (Figure 7-d).

Spatial distribution of the dependence between P and S, W or SW (Figure S5) are relatively similar to the spatial distribution

between Q and the three oceanographic variables. Highest correlation between P-S or P-W (with values around 0.3-0.4) are concentrated on the southern coast of the North Atlantic Ocean. In the case of the Mediterranean Sea, similar areas with high dependence are detected including the western coast of the Black Sean and excluding many locations along the Greek coast.


Correlation is higher when the conditional sampling is conditioned to oceanographic variables. S and W (Figure S5d) are the drivers with the highest correlation, with coefficients of around 0.6-0.7 in the north Atlantic Ocean to minimum values of 0.2

in some regions in the Mediterranean Sea.

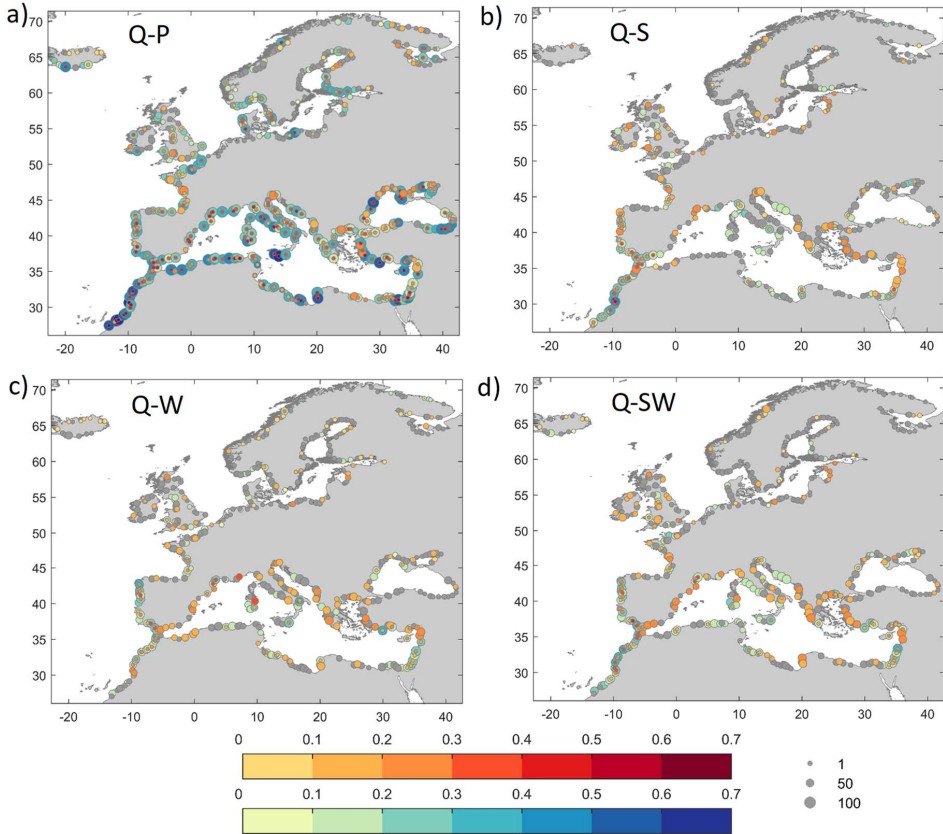

**Figure 7: Kendall's correlation coefficients (p<0.05) between multivariate extremes selected using the POT3 approach and the joint occurrences between (a) Q-P, (b) Q-S, (c) Q-W and (d) Q-SW. Correlations between compound events selected when variable 1 is**

**the dominant driver is represented on the red scale while the blue scale denotes the correlations obtained when variable 2 is the dominant driver. The size of the circles on blue scale represents the join occurrences (maximum=100). If correlation is insignificant when variable 2 is the conditioning variable, the size of the circle on red scale represents the joint occurrences. When both correlation coefficients are insignificant, the size of the grey circle represents the joint occurrences.**



### 4.2.2 Severity index

Here we define an index, based on driver severity, to be included in the characterization of the spatial patterns of compound flooding potential. The driver severity is calculated as the sum of normalized thresholds of each driver, applied in the conditional sampling (multiplied by 0.2 in the case of W). Q thresholds, which cover a wide range of values, have been categorized into 10 intervals [0-10-25-50-100-250-500-750-1000-5000->25000 m3/s] to avoid skewing the driver severity due to very high discharge magnitude in several locations. Driver severity is divided into 11 scores from 0 to 1. Figure 8d shows

the spatial distribution of the severity index (SI). Areas with highest SI are concentrated in the North Sea, the northwest of the Iberian Peninsula, the eastern coast of the Adriatic Sea, the eastern coast of the Black Sea and few locations that represent large rivers. Coastal areas with the lowest SI are mainly concentrated along the southern coast of the Mediterranean Sea and the most southern coast of the Atlantic Ocean of our study domain. The SI spatial distribution indicates that an identical SI ranking can be determined by different combinations of driver extremes. To facilitate this analysis, we classify the thresholds

of the four drivers (shown in Figure S6) into 10 clusters to define the main combinations of driver extremeness (Figure S8a). The probability of occurrence of each cluster (number of locations of the study domain represented by each cluster) associated with each SI rank (Figure S8b) provides which combinations of the four driver thresholds have an equal SI rank. For example, locations with SI equal to 0 are associated with only one cluster (represented in light green) which is defined by a combination of the lowest thresholds of Q, P and S and low W severity. On the other hand, locations with SI equal to 1 are associated with

clusters 1 and 2 (in yellow and orange, respectively) which are characterized mainly by the severity of one driver (Q or S, respectively) but also associated with clusters 3, 7 and 8 (in red, dark and light blue, respectively) with high severity of two or three drivers (Q, P and W, or Q and P, or Q and S, respectively). The spatial distribution of these clusters (Figure S8c) allows identification for each location the representative combination of the four driver thresholds.



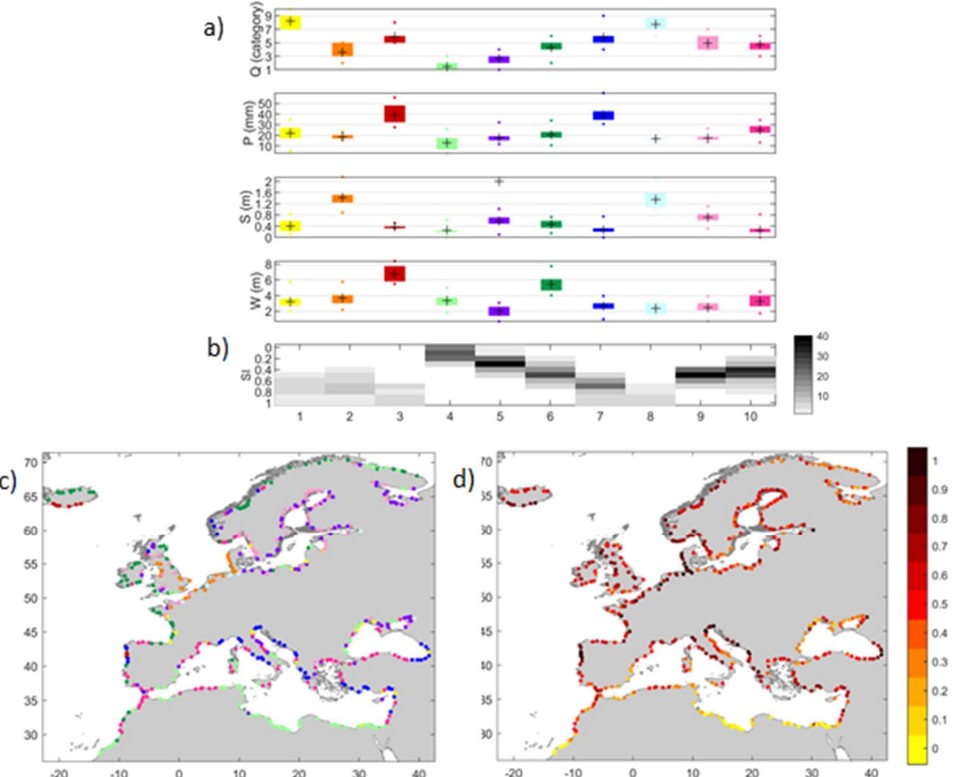

**Figure 8: a) Ten clusters defined by the combination of the thresholds of four drivers (Q, P, S, W) used in the conditional sampling (the same colour is used to represent the four thresholds by the mean, 25th, 75th, 5th and 95th within each cluster). b) Probability of occurrence of each cluster for each severity index (SI) score (grey scale shows the number of locations represented by each cluster). c) Spatial distribution of the clusters represented in the same colour as used in panel a. d) Spatial distribution of SI based on the sum of the normalized thresholds of the four drivers.**

### 4.2.3 Spatial Patterns of Compound Flooding Potential

The characterization of compound flooding potential can be summarized using the combination of two metrics: the Kendall's correlation ($\tau$) and the joint occurrence (JO) for the pairs Q-P, Q-SW, P-SW and the number of co-occurring events when all three variables are extreme JO(Q-P-SW). The two-step cascade classification method is applied to the 11-dimensional array $X_i = [\tau_1(Q\text{-}P)_i, \tau_2(P\text{-}Q)_i, JO(Q\text{-}P)_i, \tau_1(Q\text{-}SW)_i, \tau_2(SW\text{-}Q)_i, JO(Q\text{-}SW)_i, \tau_1(P\text{-}SW)_i, \tau_2(SW\text{-}P)_i, JO(P\text{-}SW)_i, JQ(Q\text{-}P\text{-}SW)_i,$



SIi], where the subscript represents the i-th grid point. Each parameter is normalized to avoid assigning different weights in the classification process. We first use the SOM algorithm to obtain a large collection of centroids (20×20=400) projected onto a 2D organized lattice that helps to analyse the dependence between the 11 parameters. The hexagonal SOM of 20x20 size of the compound flooding potential derived from the 11 metrics outlined above for the study sites is shown in Figure 9. Results

are shown in individual panels (Figures 9a-k) over the same 2D lattice for the different metrics defining the SOM centroids (the hexagons in a certain position correspond to the same map unit in each figure). Note that each Figure has a different scale. For example, we can observe that the three parameters related with the pair Q-P ($\rho1$, $\rho2$, JO; see Figure 9a-c) present a similar distribution in the lattice which means that there is a high dependence between them. Locations with the highest correlation between Q-P(P-Q) (Figures 9a and 9b) also have the highest number of joint occurrences (Figure 9c). Centroids with the

highest dependence parameters between Q-SW and P-SW are concentrated in the upper right corner of the lattice (Figure 9d-j). They are also characterized by the highest number of joint occurrences between Q-P (Figure 9c) and high-medium dependence between Q-P (Figures 9a and 9b). Accordingly, the highest joint occurrences between all three variables are also found in the upper right corner (Figure 9i). Other centroids located in the upper left side of the lattice represent locations with high dependence between P-SW (Figure 9g-h) but not between Q-SW and relatively smaller dependence between Q-P.

The severity index (Figure 9k) reveals that locations with highest driver severity do not present dependence between any pair of drivers (lower right corner of each individual panel in Figure 9). These locations are the ones represented principally by clusters with the severity determined by only one driver (clusters 1 and 2 in Figure 8a). The probability of many SOM centroids is null (Figure 9l) because a large lattice size (20x20) is required to guarantee that SOM centroids cover the multidimensional data space defined by the 11 parameters.

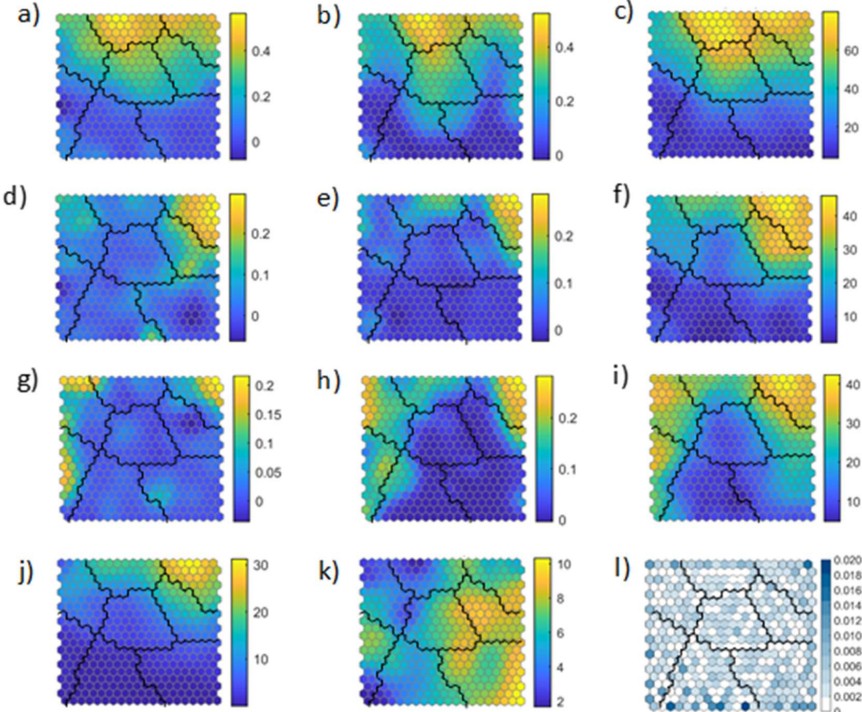

**Figure 9: SOM classification: a) τ1(Q-P), b) τ2(P-Q), c) JO(Q-P), d) τ1(Q-SW), e) τ2(SW-Q), f) JO(Q-SW), g) τ1(P-SW), h) τ2(SW-P), i) JO(P-SW), j) JQ(Q-P-SW), k) SI, l) probability of each SOM centroid.**


In the second step, we apply the KMA to find a reduced number of clusters applied to the SOM centroids. We obtain an optimal number of 8 clusters by applying the Davis-Bouldin and gap criteria. Figure 10b shows the KMA classification in 8 groups

over the SOM lattice, and highlights the similarity between KMA clusters because neighbouring centroids in the 2D lattice have similar values of the eleven parameters (see black contours of Figure 9 which delimitate the SOM centroids belonging to each KMA cluster). Figure 10a shows the mean value of the eleven parameters associated to each cluster and the variability within each group (25th, 75th, 5th, 95th percentile). The number of locations represented by each cluster is shown in Figure 10c.

The KMA classification reveals two clusters (red and pink groups) that represent locations where more compound flood events can occur. The pink group is characterized by the highest dependence (both correlation and joint occurrences) between Q-SW (SW-Q) and P-SW (SW-Q) and high dependence between Q-P (P-Q), which is reflected in the highest number joint



occurrences between the three drivers. The SI centroid presents a medium-high severity with a wide variability. The red group
represents locations with the highest dependence between Q-P (P-Q), and medium correlation, only when extremes are selected

conditioned to oceanographic drivers, and high joint occurrence between Q-SW and Q-P. It is characterized by a low-medium
severity index because most of the locations present mild meteocean conditions (represented by cluster 4 in Figure S8a). The
purple cluster follows the other two in terms of compound flooding potential. It is characterized by a high joint occurrence
between Q-SW and P-SW but lower dependence between Q-P and represents locations with a high severity index. Green and
blue clusters stand out for the high numbers of compound events resulting mainly from the combination of P-SW. The green

cluster is also characterized by significant dependence between Q-P and Q-SW at locations with low severity meteocean
climates. In contrast, the blue cluster represents locations without compound events generated by the combination of Q and
SW. The dark green cluster represents locations where only compound events from the combination of Q-P can occur. The
two remaining clusters, that represent 43 % of all study locations, are characterized by negligible compound flooding potential
and are distinguished by the severity of the drivers, with the yellow cluster representing locations with high severity index and

the light blue cluster representing locations with low severity index.



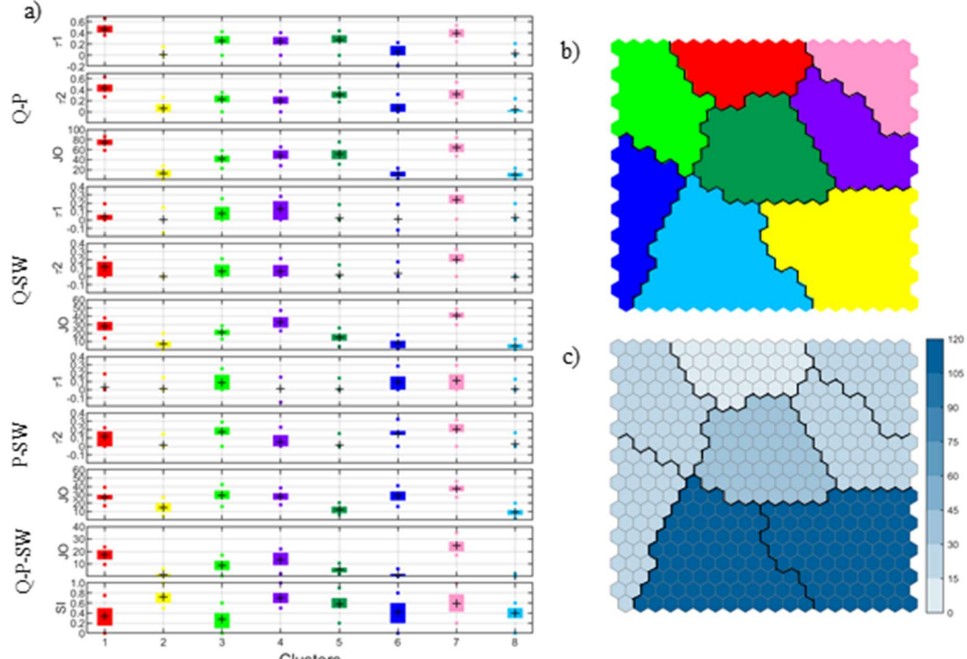


**Figure 10: KMA classification: a) Characteristics of each cluster: τ(Q-P), τ(P-Q), Co(Q-P), τ(Q-SW), τ(SW-Q), Co(Q-SW), τ(P-SW), τ (SW-P), Co(P-SW), Co(Q-P-SW), SI; b) The 8 KMA clusters over the SOM lattice; c) Number of locations represented by each KMA cluster.**

The geographical distribution of the 8 KMA clusters represents the compound flooding potential patterns across the study domain (see Figure 11). For example, the pink cluster which characterizes the pattern where the most compound events occur from the combination of the four drivers is distributed along the southern coasts of the North Atlantic Ocean, the eastern coast of France, as well as scattered locations along the northern coast and of the eastern coasts of the Mediterranean Sea. On the other hand, the red cluster is mainly localized along the most southern coast of the North Atlantic Ocean and isolated locations

in the Mediterranean and Black Sea; recall that this cluster represents locations with low driver severity. Other locations identified with significant compound flooding potential, including along the western coast of France and UK or the north-eastern coast of the Mediterranean Sea, are part of the purple cluster.


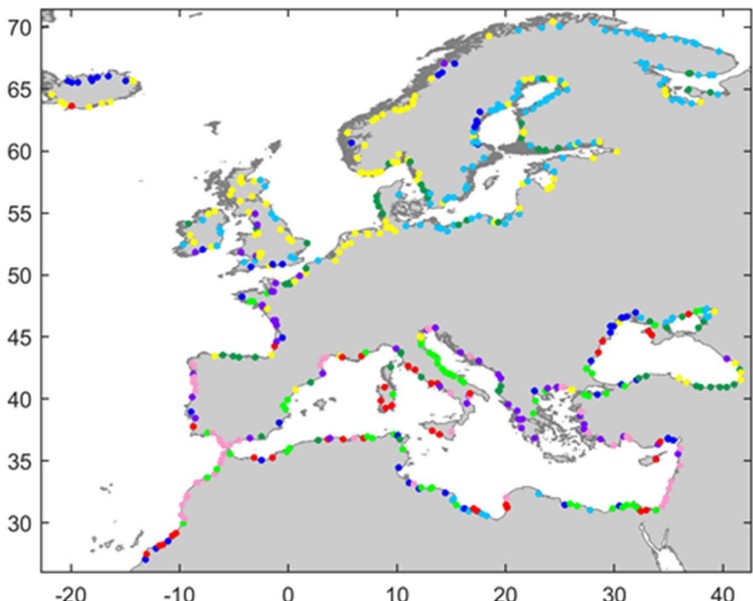

**Figure 11: Compound flooding potential patterns identified in this study as highlighted by the spatial distribution of the 8 KMA**
**clusters. Each KMA cluster is represented in the same colour as used in Figure 10.**

## 5 Discussion

In this paper, we have analysed the compound flooding potential arising from pluvial, fluvial and oceanographic sources. The
assessment is based on a bivariate analysis of the dependence between drivers (P, Q, S and W) using a two-sided conditional
sampling. Our first objective is focused on the analysis of the sensitivity of the results to several factors that have been applied
indiscriminately in previous studies with the purpose of identifying compound events and characterizing compound flooding
potential.

First, we apply AM and POT sampling approaches to analyse how the choice of these approaches affects the computed
dependency between variables. It is noteworthy that our results show that a lower statistical dependence is obtained when
using the POT method, which is in agreement with Ward et al. (2018), yet a broad consensus has emerged in favour of the use
of the POT approach for identifying extreme events (Mazas et al., 2014, Coles, 2001). AM can potentially disregard
information on extremes in the remaining data from using only one data point per year (Méndez et al., 2016), or select events





that are not extreme, as we have observed in several of our study sites. However, the spatial patterns in both approaches are similar and comparable areas are identified as hotspots with relatively higher dependence.

In our second sensitivity analysis, we found that the choice of the time window used has almost a negligible effect on the computed correlation coefficients (at least for the time windows in excess of the 3 days used here). The higher probability of finding more severe events using a longer Δt has only be reflected in a higher correlation in few locations. However, it can result in a lower number of compound events when both drivers are extreme (i.e. less joint occurrences).

  In the third sensitivity analysis, we investigated differences in the characterization of the dependence between drivers using

different metrics. We found that the correlations obtained are always higher when using the Spearman's rank correlation coefficient as compared to the Kendall's coefficient, but they are unequivocally related, and equivalent spatial distributions are obtained irrespective of the choice of the correlation coefficient. Regarding joint occurrences and tail dependence, we found that both provide comparable quantifications of the dependence between driver variables when both variables are extreme. Moreover, the concept of joint occurrences provides a better measure of the compound flooding potential because it

applies a declustering method to select independent events.

  The last factor we assessed is the definition of wave-driven contribution to the sea level when using the sum of the S and W components. Any differences in correlation emerging as a result of the definition of wave-driven contribution to sea level seem to be explained by a higher dependence between surge and the simplified wave-driven sea level (20% of W) than surge and setup based also on the wave period (see Figure S7). Although the same beach slope is considered in all study locations, our

results showed that the combination of W and Tp in the estimation of setup influences more the selection of compound events conditioned to sea level than the semi-empirical formulation itself.

  Our second objective was focused on estimating the spatial distribution of compound flooding potential considering the drivers Q, P, S, W, and SW. We observed significant differences in the dependence between the pairs of drivers and even for one individual pair depending on which driver is employed as the dominant one in the selection of the compound events. We find

that the correlation coefficient and joint occurrences are not always positively related. Therefore, we considered that combinations of both metrics provide complementary information about the type of compound events and represents different flooding mechanisms (Wahl et al., 2015). The joint occurrence only characterizes compound events when both drivers are extreme. On the other hand, correlation coefficients characterize those compound events generated when one of the drivers is extreme but not necessarily the other, providing information about the relative severity of the secondary driver.

Regarding a comparison with previous global and regional studies of compound flooding potential in the study domain, the hotspots we have identified on the coasts of Portugal, the Strait of Gibraltar and Morocco have also been detected in Couasnon et al. (2020). However, although we found a higher number of joint Q-S occurrences on the south-west and west coasts of the UK than on the east coast, as previously noted by Hendry et al. (2019), the number of co-occurrences is lower in our analysis, as is also the case around the coast of Ireland. Similar high joint occurrences between Q-S in the northern and eastern

Mediterranean coasts and in the coast of Tunisia are found, in accordance with Couasnon et al. (2020). We do not observe a predominance of higher correlation when compound events are conditioned to Q as in Coausnon et al. (2020) found. However,





differences in the correlation between the two conditional samples are found between P and S, W or SW. As pointed out by Hendry et al. (2019), storms that generate high precipitation are different to the ones that generate high storm surges. Specifically, heavy precipitation and extreme surges are driven by deep low-pressure systems, while intense rainfall can also

be caused by convection without intense cyclonic activity (Bevacqua et al., 2019). Therefore, there is higher probability of compound events when S or W are the dominant variables (generated by extratropical storms) than when compound events are conditioned to P (convective storms). This effect seems to be less perceptible between river discharge and oceanographic variables because other climatic and non-climatic factors affect the fluvial source driver (Bevacqua et al., 2020), as for example, land use characteristics or snowmelt, evaporation, and accumulated precipitation over previous weeks or months.

Bevacqua et al., 2019 found the lowest joint return periods due to high dependence between P-S were concentrated along the Atlantic coast and in the Mediterranean Sea (particularly in the regions of the Gulf of Valencia (Spain), the northwest Algeria, the Gulf of Lion (France), the Adriatic coast of Balkan Peninsula, the Aegean coast, southern Turkey and the Levante region). Even though we did not calculate return periods, our results suggest similar areas of higher dependency between P-S. We find a distinct pattern between southern and northern European coasts with more joint occurrences between S-W, especially over

the Irish Sea, English Channel and south coasts of the North Sea and Baltic Sea in line with the results of Petroliagkis (2018). Similar regional patterns of dependence between S-W as we find here were reported by Marcos et al. (2019), but we find some additional local areas with strong dependence when both drivers are extreme (characterized by the statistic χ or joint occurrence) such as the western coast of the Iberian Peninsula and certain areas in the Mediterranean Sea, perhaps because we used a higher Δt than in Marcos et al. (2019).

We apply a two-step clustering method to synthesize the high dimensional results of the bivariate characterization of compound flooding potential from the four sources. First, the SOM algorithm allows us to analyse the multivariate dependence between the correlation coefficients and the joint occurrences between the pairs of drivers Q-P, Q-SW and P-SW, enabling the establishment of the degree of contribution of each driver combination to compound flooding in the study area. Moreover, the method also distinguishes if the identified compound events are more likely to occur when both drivers are extreme or when

only one driver is extreme. With the second step of the clustering method, we identify a reduced number of types of compound events based on the contribution of each driver combination and the driver severity. The spatial distribution of these types of compound events reveals spatial patterns of compound flooding potential. These patterns allow us to discern locations with the highest overall compound flooding potential and the associated contributions of each pair of drivers. Additionally, we introduce a severity index to distinguish between locations with similar compound flooding potential (from the dependence

perspective) but very different driver severity.

The main limitation of our study is the identification of the compound events based on drivers. None of the contributing variables have to be necessarily extreme to create a compound flooding event. Therefore, the selection of compound events should ideally be based on an impact function or a risk function that accounts for exposure and vulnerability. However, this function is usually unknown and difficult to derive, especially in regional and global studies. An intermediate approach could

be based on the selection of the compound events in the extreme water levels generated by the interaction between



oceanographic drivers and riverine drivers. The amplification of the flooding impact has been identified at the local scale (van den Hurk et al. 2015, Kumbier et al. 2018), and recently at global scale (Eilander et al. 2020) using a global coupled river-coast flood model framework (Ikeuchi et al., 2017).

**6 Conclusions**

In this paper we have evaluated the compound flooding potential arising from the combination of precipitation, river discharge, storm surge and waves along the coasts of the eastern North Atlantic Ocean, Mediterranean Sea and Black Sea. The paper provides two advances. First, we performed a series of sensitivity analyses to establish how methodological choices affect the identification of compound flood events. Specifically, we investigated: 1) the sampling method, 2) the time window used to
match events in the two-way sampling, 3) the use of typical metrics applied in the evaluation of the dependence between drivers, and 4) the definition of the wave-driven sea level contribution. Among these, the sampling approach produces the highest differences in the quantification of the compound flooding potential. However, none of these factors analyzed here introduces significant differences in the spatial distribution of the compound flooding potential which means that similar locations where compound flooding is more likely to occur are identified.

The second major advance forwarded by our work comprises a new regional characterization of compound flood potential using a methodology which aggregates the bivariate dependence between driver combinations. This multivariate characterization reveals three main locations with high compound flooding potential: the southern coast of the North Atlantic, the western coasts of France and UK and the northern coast of the Mediterranean Sea. These locations are characterized by compound events that arise from the combination of the four drivers, albeit with differences related to the driver severity, the
contribution of each pair of drivers and the predominance of compound events when drivers are extreme. Other locations of relatively high compound flood potential include the eastern coast of Italy and southern Mediterranean Sea, where compound flooding is mainly driven by combinations of P-SW.

This regional quantitative assessment of multivariate compound flooding potential can be considered as a pre-diagnostic tool for coastal management. Results provide information about which areas are more predisposed to experience compound
flooding. In addition, this multivariate flooding potential classification identifies the relevant drivers of coastal compound flooding at each location which can assist to decide the most appropriate methodological approach to perform high-resolution hydrodynamic and impact modelling.




**Data availability**

ERA5 (https://doi.org/10.24381/cds.adbb2d47), GloFAS-ERA5 (https://doi.org/10.24381/cds.a4fdd6b9) and CoDEC
(https://doi.org/10.24381/cds.8c59054f) data are available on the Copernicus Climate Change Service (C3S) Climate
Data Store

**Author contribution**

IH and TW conceived the project and supervised the work. PC performed all the analyses, and wrote the first draft of the
manuscript. IH, TW, AH provided advice on the methodology and on data analysis, discussed the results and contributed to
the writing. SD and RN contributed to the interpretation of the results and reviewed the manuscript.

**Competing interests**

The authors declare that they have no conflict of interest

**Acknowledgement**

P.C. and I.D.H.'s time on this research has been supported by the UK NERC grant CHANCE (grant no. NE/S010262/1). This
material is based upon work supported by the National Science Foundation (NERC-NSF joint funding opportunity) under NSF
Grant Number 1929382 (T.W and A.N.).

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
