# Peer review of "Regional analysis of multivariate compound coastal flooding potential around Europe and environs: sensitivity analysis and spatial patterns"

_Natural Hazards and Earth System Sciences, 2021_

## Author Comment (AC1)

Dear Editor,

We would like to thank the reviewers for their positive, thoughtful and constructive comments. We have revised the text carefully and amended the questions raised. We outlined the changes made point by point below (answer in blue and new in the manuscript in red).

Regards,

The authors

**REVIEWER 1**

Review of the paper "Regional analysis of multivariate compound flooding potential: sensitivity analysis and spatial patterns" submitted to NHESS by Camus et al.

This is a well-written manuscript that analyses the compound flooding potential in Europe considering 4 drivers: precipitation, river discharge, storm surge and waves. To my knowledge, it is the first study that analyses these 4 drivers altogether at European scale, therefore providing meaningful insight about compound flooding potential. This study also provides a uselful sensitivity assessment to different choices that can be made when analysing compound events, such as sampling method, time window, etc. Lastly, they derive a severity index that combines the effects of all drivers into one index to explain potential coastal flooding, which is later analyzed with self organizing maps and kmeans cluster analysis to identify hotspots of potential flooding.

I recommend its publication after addressing the comments below (minor revision).

Conceptual aspects:

1) Line 17 "with homegeneous forcing" (also in line 97). I think the term "homogeneous" is not correct here. Modern reanalysis are subject to temporal inhomogeneities due to increasing amount and type of assimilated data over time. I suggest to replace it by "with the same forcing" or "with coherent forcing", I would also remove the "temporally consistent" part in line 105.

We have changed "homogeneous" to "coherent".

2) Line 74-83. It should also be mentioned that, if information about the impact is available, another option is conditional sampling on the impact variable, ie analyse the behaviour of the drivers/predictors when the impact is extreme.

We mention this option of sampling the extreme compound events based on an impact function in the discussion (lines 636-644). The introduction contains mainly the approaches followed in other regional and global studies, where such impact metrics are typically not available.

3) Line 180: "ignoring the astronomical tidal component of sea level, as it is deterministic". I am not sure this is a valid argument. It is true that astronomical tides are deterministic but the timing in which large tide might occur respective of, for example, large storm surges is not. I suggest removing the deterministic part or change it to "which is deterministic" or similar.

It is true that the phase of the high tide in relation with the highest daily storm surge is not deterministic but the variable itself can be considered deterministic (e.g., completely predictable). However, we are primarily interested here only in the analysis of climate drivers, especially the ones related with the same weather systems that can generate compound events. That is the reason why we consider only the storm surge and ignore the tidal component, which itself is not related in any way to weather systems.

As you suggest, we have changed "as it is deterministic" to "which is deterministic" and include this comment "(we note that for actual flood risk assessments the timing of tidal levels with the other drivers is important but this is beyond the scope of this analysis)".

4) Line 185-186 I am a bit confused about these pairs. I thought that waves were characterized in terms of SW (simplified wave contribution to water level), and WL (wave set up including the effect of Tp + surge). First, why do you consider surge together with wave set up when is calculated considereing Tp, but then wave set up and surge are separate when using the simpler formula? Then why WL does not appear as part of the pair of driver combinations analysed? Maybe I got something wrong but I suggest clarifying this in the text.

Yes, it is true that the description of the sea level driver was a bit confusing and needed clarification; thanks for pointing this out. First, we consider waves as independent driver (W = significant wave height) and then also as the linear combination of storm surge (S) and wave-driven sea level component. For the later, we use the following two definitions, which are often used in the literature:

1) SW which is S + simplified wave contribution, 0.2W; and

2) WL which is S + wave setup, which in turn is based on W and Tp.

The analysis is mainly shown using SW (section 4.2) and only the comparison of dependence between P or Q and sea level using SW and WL is shown section 4.1.4. We have introduced the pair of drivers P-WL [WL-P] and P-WL [WL-P] in the list of paired driver combinations (Lines 180-187). We have also changed the description of the two definitions of sea level in Lines 267-274, as follows.

*Lines 184-192:*

*"We do this considering two definitions of the wave-driven sea level component: 1) a simplified definition for the wave contribution to sea level given as 0.2W (e.g., Vousdoukas et al., 2017) , and 2) through the use of more sophisticated semi-empirical formulations (e.g., Vitousek et al., 2017), that also takes into account the wave period (Tp) to calculate Setup. We  refer to the resulting variables after combining the surge and wave contribution respectively as SW (sum of S and 0.2W) and  WL (sum of S and semi-empirical Setup)  and compare results of compound*

*flooding potential using both definitions. The seven paired driver combinations we consider here are, therefore: 1) Q-P [P-Q]; 2) Q-S [S-Q]; 3) Q-W [W-Q]; 4) Q- SW [SW-Q] or Q-WL [WL-Q]; 5) P-S [S-Q]; 6) P-W [W-P], and7) P-SW [SW-P] or P-WL [WL-P].*

*Lines 276-284:*

*The setup contribution is defined  with different levels of sophistication. Wave setup has been approximated as the significant wave height multiplied by 0.2 (Vousdoukas et al., 2018, Bevacqua et al., 2019, Marcos et al., 2019) or by applying the Stockdon formulation (Stockdon et al., 2006) with different parameterizations (Vitousek et al., 2017, Rueda et al., 2017, Melet et al., 2018). The wave setup contribution to the total water level is very sensitive to this parameterization (Aucan et al., 2018). Following Vitousek et al. (2017), we used the Stockdon formulation for dissipative beaches (Eq. 1), which is known to provide similar results as using a beach slope of 0.02 (~50% of the world's beaches have slopes smaller than 0.02, Aucan et al., 2018), ~~Furthermore, in the definition of the sea level as the sum of S and 0.2W, here called SW, we have also applied the Stockdon formulation for wave setup for dissipative beaches (as Vitousek et al., 2017) which is known to provide similar results as using a beach slope of 0.02 (~50% of the world's beaches have slopes smaller than 0.02, Aucan et al., 2018), here termed water level (WL), as the sum of S and Setup.~~ The two variables we use here to represent the sea level are: 1) SW as the sum of S and setup (given as 0.2W), and 2) WL as the sum of S and the setup calculated using the Stockdon formulation.*

5) Line 204-205: I suggest listing threshold as the first parameter introduced in POT. Then I would keep the other factor more general, such us "criteria used to select independent storms" (it is not always just one time window between peaks, alternative or complementary times can be used to determine if two storms can be considered independent, such as the time below the threshold between storms)

*This is a good point, we have changed the order of the parameters to emphasize that threshold in the main one using POT and the definition of independent event (Lines 206-209).*

*On the other hand, the POT approach increases the amount of selected extreme events but introduces two parameters in the selection process: 1) the threshold; 2) the definition of independent events established by a minimum time between peaks or below the threshold.*

6) Section 4.1.1 What is the time window used to assess the sampling sensitivity?

*We used a time window of ± 3 days. It is specified in Line 305.*

7) Figure 3. As noted by the authors, it is expected that the number of co-ocurring events are larger for POT6 than POT3, and POT3 than AM, as the number of total events differs for each sampling method. I suggest to show a normalized number of co-ocurring events, which might provide more interesting information.

*As you suggest, we have recreated Figure 3 (see below) with normalized number of co-occurring events. We think this visualization distorts the comparison of ranges of joint*

occurrences between the different methods. The main message we want to deliver with this figure is the linear relationship between the results using the three methods and we think that the original version of the figure with the slopes marked is more appropriate for this purpose, hence we have not made this change.

[Figure]

Figure 1. Scatter plots of the normalized joint occurrences of AM against POT3 and POT3 against POT6.

8) Line 447. Why is W multiplied by 0.2 if afterwards is normalized?

It is true that it is the same. It is just to stress that the contribution of waves is not just the magnitude of the significant wave height and 0.2Hs is just a simplification.

9) Section 5. I acknowledge the authors provide an extended discussion of many of the results. However, I would discuss a bit more the results obtained, such as why correlation with AM is larger than with POT. Could it be because the sample is shorter? Also, why the correlation is lower when using a wave set up formulation that includes Tp? Influence of remote swells that are not correlated with local storms?

Regarding the first comment about higher correlation when using AM, we have included the following in the discussion section (Lines 584-587):

*"The larger correlation coefficient derived when using the AM approach might be due to a higher tendency that annual peaks of both drivers co-occurred, when the dependence between them is significant, while the POT method selects more combinations where drivers are less extreme, which in turn is reflected in a lower correlation coefficient".*

Regarding a lower correlation when wave setup formulation is used, there is a short explanation about this effect in Lines 586-591, which is limited to the comparison of correlation between storm surge and the two definitions of wave-driven sea level. To examine this further, we have also analysed the Hs and Tp values of the selected SW or WL events (see Figures below). It seems that in locations in the most southern North Atlantic coastlines, there is more probability of longer Tp which could correspond to swells. We have added a comment about this to the manuscript, as follows.

*"The last factor we assessed is the definition of wave-driven contribution to the sea level when using the sum of the S and W components. Any differences in correlation emerging as a result of the definition of wave-driven contribution to sea level seem to be explained by a higher dependence between surge and the simplified wave-driven sea level (20% of W) than surge and setup based also on the wave period (see Figure S7); this could be due to the influence of swells."*

[Figure]

Figure 2. 2D-Histograms of Hs and Tp corresponding to the selected events using POT when SW or WL variables are dominant at location Lon=-8.7671°; Lat=41.3599°.

[Figure]

Figure 3. 2D-Histograms of Hs and Tp corresponding to the selected events using POT when SW or WL variables are dominant at location Lon=-8.3569°; Lat=33.3179°.

Formal aspects:

Line 48: add space ; and space between Couasnon and Ridder citations.

Done.

Line 53: This paragraph is quite long. I suggest dividing it in two, for example just before "High-dimensional systems can be modelled using…" in line 60.

Done.

Figure 1 is hard to see (specially panel a). I suggest making the figure larger.

It has been enlarged, as much as possible.

Line 182: I suggest adding in the manuscript the exact formula used to calculate wave set up.

Done.

Figure 2. Figure is too small, and I also suggest making the legends smaller so there is not overlap with the histograms bars.

Done.

Figure 6. I suggest to clarifying the difference of what is substracted from what. Perhaps add subpanels titles.

We have changed the figure caption to:

*Figure 6: Differences in the Kendall's correlation coefficient (p<0.05)  between [SW-Q]Q and [WL-Q] (left panel) and between [SW-P] and [WL-P] P  (right panel), when SW or WL are the dominant variables, respectively.*

Figure 7. Isn't there overlapping of red and blue colours when the joint occurences (expressed as circle size) are similar for both combinations of pairs?

No, when there are red and blue circles (both correlation coefficients significant), the red circle does not change the size and the blue circles is always bigger with a size function of joint occurrences.

Section 4.2.2. I believe Figure S8 should be Figure 8 (line 455, 457 and 462)

Yes. Several panels of Figure 8 were in Supplementary Material in a previous version.

Line 606: remove "found" after Coausnon et al. (2020)

Done.

**REVIEWER 2**

This study on compound floods in Europe has two main aspects, one is investigating how sensitive are the compound flood estimates to the choice of method, and second is finding a synthetic measure of their severity. In both cases the paper provides meaningful results and I think will be an important work of reference. The manuscript is rather clear and concise, even though the presented findings are quite technical and not easy to synthesize. Overall I didn't really see any major issues with the study beyond those indicated by Referee #1.

The only thing I really found missing is that no reference is made to potential bias or overarching inaccuracy in the models in reproducing compound events, method used notwithstanding. As Ganguli et al. (2020) and Paprotny et al. (2020) have shown, the dependency between compound flood drivers produced by models differ considerably from dependencies computed from observations. For north-western Europe, where compound events are potentially most frequent and most severe there was a positive bias in the models used in the studies. I think the authors should comment on this aspect in the discussion in context of their calculation of the severity index and clusters.

*Thank you for this. We have introduced this comment in the discussion (Lines 660-672) as follows, and added the references you suggest:*

*"Although compound flooding drivers have been found to be generally captured well in different reanalysis and hindcast products (Paprotny et al., 2020), differences in the strength of dependence derived from observations and models can vary spatially and across different variables, which is more evident when regional climate models are used, even after bias corrections have been made (Ganguli et al., 2020). In addition, Paprotny et al. (2020) also detected false positive and large compound floods in observation that were missed in the modelled data products. We therefore acknowledge that model biases might mischaracterize absolute values of dependence in some cases. However, the conclusions we draw from our results regarding the sensitivity analysis are not likely to be altered and the relative importance of drivers and spatial patterns would also likely be less (or not at all) affected".*

Below I list a few minor textual points. I'm looking forward to the authors' revision of the paper.

Minor points:

L86: Paprotny et al. (2020) considered four drivers as well.

*We have differentiated any study that has analysed the dependence between waves and the other three drivers independently. This is the reason why we introduce Paprotny et al. (2020) with the Bevacqua reference. In both studies, the wave-driven sea level component has been included in the sea level directly (by combining wave height with storm surge and/or astronomical tide) without analysing the correlation with the other drivers separately. Also, we have removed the reference of Hendry et al., (2021) because it is under review.*

*We have made the following changes between Lines 84-93:*

*In the compound flooding studies summarised above, it is evident that a wide range of different statistical approaches have been used to define compound flooding potential, usually caused*

*by the combination of two drivers. To date, no study has yet analysed the dependence between the four potential drivers of flooding in coastal regions independently for each pair combination.*  *Ridder et al. (2021) have identified hotspots of compound events that potentially cause high-impact floods related to wet conditions based on the joint occurrence of multiple hazards pairs (precipitation, wind, hail, streamflow and storm surge). In other studies, the wave component has typically been included in the sea level directly (by combining wave height with storm surge and/or astronomical tide) without analysing the correlation with the other drivers (Bevacqua et al., 2019, Paprotny et al., 2020).*

L180-185: I concur with reviewer #1 that this paragraph is very unclear as to how storm surge and waves are combined, and reading 3.1.4 doesn't help much, until actually reading the results. I suggest to use a single letter to define the combined surge-wave height (e.g. H or L) and then differentiate the two methods for computing the joint height as e.g. H1 and H2, so that the relevant pairs are referred as e.g. Q-H [H-Q].

Please, see our answer above to comment 4 of Reviewer 1.

L226: "between 1 and 10 days or 1 and 3 days". I think would be clearer to write "of ±10 and ±3 days".

Done.

L485 & L510: in "JQ(Q-P-SW)I" it should be "JO", I guess?

Yes, it has been changed.

L540: the 'large' labels of the vertical axis, indicating the pairs do not clearly connect with their variants in smaller fonts, and further are not aligned with the caption which has some undefined abbreviations e.g. "Co". Please check that and synchronize the naming with the text.

The figure caption has been changed and the labels of the figure were better aligned. We tried to introduced large labels in the smaller fonts but there was not enough space to do this.

Supplement: Figure S7's caption indicates that Kendall's correlation is used, yet the panels are labelled with "rho", suggesting Spearman's correlation. Please check what data is actually shown.

We have eliminated the Pearson correlation coefficient from all of the figures. It was the correlation between the data that compare and show in the plot.

References:

Ganguli P., Paprotny D., Hasan M., Güntner A., Merz B. 2020. Projected changes in compound flood hazard from riverine and coastal floods in Northwestern Europe. Earth's Future 8(11):e2020EF001752, doi:10.1029/2020EF001752

Paprotny D., Vousdoukas M.I., Morales Nápoles O., Jonkman S.N., Feyen L. 2020. Pan-European hydrodynamic models and their ability to identify compound floods. Natural Hazards, 101(3), 933–957, doi:10.1007/s11069-020-03902-3